# Benefits of Exercise and Astaxanthin Supplementation: Are There Additive or Synergistic Effects?

**DOI:** 10.3390/antiox10060870

**Published:** 2021-05-28

**Authors:** Leandro Kansuke Oharomari, Mitsushi J. Ikemoto, Dong Joo Hwang, Hikaru Koizumi, Hideaki Soya

**Affiliations:** 1Laboratory of Exercise Biochemistry and Neuroendocrinology, University of Tsukuba, Tsukuba 305-8574, Japan; lkansuke@hotmail.com (L.K.O.); dongzoo87@gmail.com (D.J.H.); aoxora333@gmail.com (H.K.); 2Sport Neuroscience Division, Advanced Research Initiative for Human High Performance (ARIHHP), Faculty of Health and Sports Sciences, University of Tsukuba, Tsukuba 305-8574, Japan; 3Molecular Composite Physiology Research Group, Health and Medical Research Institute, National Institute of Advanced Industrial Science and Technology (AIST), Tsukuba 305-8566, Japan; m.ikemoto@aist.go.jp; 4Graduate School of Science, Toho University, Funabashi, Chiba 274-8510, Japan

**Keywords:** astaxanthin, exercise, health, additive effect, synergic effect, performance, cognition

## Abstract

A healthy lifestyle is essential for maintaining physical and mental health. Health promotion, with a particular emphasis on regular exercise and a healthy diet, is one of the emerging trends in healthcare. However, the way in which exercise training and nutrients from dietary intake interact with each other to promote additive, synergistic, or antagonistic effects on physiological functions leading to health promotion, and the possible underlying biomolecular mechanisms of such interactions, remain poorly understood. A healthy diet is characterized by a high intake of various bioactive compounds usually found in natural, organic, and fresh foodstuffs. Among these bioactive compounds, astaxanthin (ASX), a red carotenoid pigment especially found in seafood, has been recognized in the scientific literature as a potential nutraceutical due to its antioxidant, anti-inflammatory, and neurotrophic properties. Therefore, scientists are currently exploring whether this promising nutrient can increase the well-known benefits of exercise on health and disease prevention. Hence, the present review aimed to compile and summarize the current scientific evidence for ASX supplementation in association with exercise regimes, and evaluate the additive or synergistic effects on physiological functions and health when both interventions are combined. The new insights into the combination paradigm of exercise and nutritional supplementation raise awareness of the importance of integrative studies, particularly for future research directions in the field of health and sports nutrition science.

## 1. Introduction

A physically active lifestyle and healthy dietary habits are two key modifiable environmental factors that play a critical role in exercise performance and disease prevention [1,2,3,4]. The majority of studies to date have focused on one intervention alone (e.g., exercise regimes or nutritional modulation) to clarify, separately, the effects and molecular mechanisms of each variable on human physiological outcomes [5,6]. Other studies compare which intervention is the best for improving some physiological parameters, e.g., adipose tissue accumulation [7]. More recently, a smarter approach has been applied. Rather than pitting nutrition and exercise against each other, scientists are trying to determine how these two research branches combined can interact additively or synergistically in their effect on health and performance.

Nutrition and sport science are huge areas of research. In terms of exercise training, different exercise protocols and methods (e.g., intensity or duration) have been tested to explore their specific effects on the clinical outcomes of conditions such as cardiovascular diseases [8], intraocular pressure [9], and immune response [10]. On the other hand, different diets [11] and nutritional supplements have been tested to evaluate their effects as prophylactic therapies [12] or as ergogenic tools for increasing exercise performance [13]. Among nutritional supplements, astaxanthin (ASX) is one of the nutraceuticals that are gaining attention in scientific literature, especially regarding the fields of muscle recovery and fatigue, aging, and, more recently, cognitive function [14,15,16]. It is worthy of note that both astaxanthin [15,16] and exercise [10] have independently shown immunomodulatory effects, an important aspect for the prevention of inflammatory diseases such as type 2 diabetes, stroke, and cancer. Therefore, concomitant ASX intake and exercise training presents a promising approach to determining how exercise and nutritional intervention can interact with each other to produce positive effects on health conditions or illnesses.

Hence, this review aimed to compile and summarize the current scientific knowledge of exercise training and ASX supplementation, focusing on its additive or synergistic effects. To that end, in 2021, a general and simple keyword search, “astaxanthin AND exercise”, was used in the PubMed online database. No filters were used in order to find the largest number of studies possible. As a result, 42 studies from 2003 to 2021 were found. The titles and abstracts of all papers were read to check whether they fit this review’s criterion. Then, 20 English-language papers which were published in peer-review scientific journals, and which discussed ASX and exercise interventions, were selected. Studies that used a mix of nutritional supplements were excluded from the analysis, since it is not possible to determine whether the outcomes described were due to ASX or other nutrients.

## 2. Astaxanthin

To date, more than 800 carotenoids have been discovered. Among them, ASX has caught the interest of the scientific community with regards to its potential biological effects, due to the features promoted by its optical symmetry, and to the hydroxyl plus keto endings on both ionone rings (Figure 1) [17]. ASX is categorized as a xanthophyll carotenoid, which sets it apart from its counterpart group, hydrocarbon carotenoids, due to its chemical structure, which includes the presence of oxygen atoms in its extremities. These hydroxylated derivatives are responsible for the antioxidant effects found in xanthophyll carotenoids. They scavenge reactive oxygen species (ROS), especially singlet oxygen. At the same time, they prevent oxidation in the conjugated backbone C = C bond, thereby preventing lipid membrane peroxidation [18]. Indeed, two in-vitro reports showed that ASX’s antioxidant effect is a hundredfold higher than that of other common nutrients such as α-tocopherol [19,20].

From an evolutionary point of view, another interesting characteristic of carotenoids regards their molecular length, with approximately 10–12 double conjugated bonds in their backbone, which is similar to the length of a bilayer phospholipid cell membrane [21]. Dipolar carotenoids like xanthophylls (e.g., lutein, zeaxanthin, β-cryptoxanthin, or ASX) are inserted in cell-membrane interfaces in a perpendicular orientation (“molecular rivets”, Figure 2). This particular position gives stability and protection to bacterial cell membranes, leading to an evolutionary advantage for these microorganisms, which have existed since the early ages of the Earth. It also exposes hydroxylated derivates to the cytosol and outside cell membrane, providing excellent antioxidant localization. This membrane biophysical property can also partially explain the several roles that carotenoids play in mammals (e.g., in the retina and brain) [22,23].

Although animals cannot synthesize ASX, its intake through microalgae is responsible for the dark red color found in many forms of marine life (e.g., fish eggs, shrimp, lobster, salmon, etc.). It is worthy of note that ASX sources may vary between artificial, synthetic sources, microalgae, crustacean by-products, aquaculture, or wild seafood. The source of ASX can contribute to its differing associations with fatty acids, proteins, and lipoproteins. The source can also influence ASX’s hydroxyl group esterification (non-ester, monoester, diester), and provide different ratios of ASX’s geometric, configurational, and optical isomers (3R,3′R; 3S,3′S; 3R,3′S (meso)) [16]. In the twentieth century, synthetic ASX production was allowed on an industrialized, large scale, reducing its cost, and making the synthetic form of ASX the most popular choice in the aquaculture industry [16]. However, in recent years the development of natural ASX production from *Haematococcus pluvialis* has significantly improved [24]. Natural *Haematococcus* ASX has three differences from synthetic ASX: (i) it differs in esterification (natural: monoester, diester; synthetic: non-ester), (ii) it differs in stereochemistry (natural: 100% 3S,3′S; artificial: 25% 3S,3′S), and (iii) it provides an additional mix of other natural carotenoids [25]. In 2013, an original paper reported stronger antioxidant effects of natural ASX compared to synthetic ASX [25]. Its source (wild vs. aquaculture fish) also seems to cause ASX to be metabolized differently in humans [26]. Taken together, these results suggest that ASX studies should take into account and describe the source of the ASX used.

In 2019, ASX’s popularity in cosmetic products and as a common additive in the food industry forced the European Commission to issue a report regarding its safety. The Panel on Nutrition, Novel Foods, and Food Allergens concluded that an intake of 8 mg per day plus the possible ingestion of ASX via daily diet is safe for adult humans [27]. However, a review claimed that the daily dosage set by the panel was based mainly on animal studies using artificial ASX supplements. The review’s final message called attention to the necessity of more human clinical trials, both for natural and artificial ASX [28]. It is worthy of note that an in vitro study found ASX toxicity by significantly decreasing gap junctional intercellular communications, an important intercellular channel that performs many physiological functions in human tissues [29]. Furthermore, to date, there is no systematic review, followed by meta-analyses or phase III or IV clinical trials, that establishes the efficacy and safety of ASX treatment as a disease-specific targeting drug. Therefore, ASX dosage safety is still under debate.

Animal and in vitro studies have provided pieces of promising evidence for the possible positive biological effects of this compound. Following this evidence in basic studies, exercise has also shown several beneficial effects on rodent models. Currently, there is no strong evidence in human clinical trials regarding the biological effects of ASX. This is because the effects of ASX have been diluted and tested under many different conditions, making it difficult to compile the data in one strong report such as a systematic review followed by meta-analyses. For instance, ASX has been tested on ophthalmological diseases [30], for exercise performance [31], and for skin and cosmetic benefits [32]. However, all of these recent and extensive reviews of the scientific literature concluded that more homogeneous, long-term, carefully managed double-blind randomized-controlled trials are warranted in order to establish the benefits and safety of ASX supplementation in each respective circumstance. Therefore, the hypothesis that ASX can interact with exercise regimes was raised, especially using animal models instead of human research, as shown in the next section.

## 3. Astaxanthin and Exercise: Animal Studies

Among papers on animal research, there are ten rodent model studies relevant to the purpose of this review (Table 1). Five of these evaluated the effects of ASX supplementation on exercise performance, and all five showed positive effects [33,34,35,36,37]. Another five studies evaluated the effects of ASX on health-related parameters [38,39,40,41,42], of which two were related to cognitive function [41,42]. The animal studies differed in rodent age (4–14 weeks old) and model (Wistar rats, ddY mice, ICR mice, and C57BL/6J mice).

All studies used ASX derived from natural sources, usually provided by the same company. The duration of astaxanthin supplementation was, on average, from two to six weeks. Unfortunately, a comparison of ASX dosage between studies is difficult since most studies lack information regarding daily food intake and animal body weight, which is essential information to calculate the intake (mg/kg. b.w./day). Seven studies administered the ASX supplement by mixing it into the diet (0.02–0.5% *w*/*w*) while three studies provided ASX by gavage (1–30 mg/kg b.w./day). Despite the heterogeneity between studies, all of them revealed promising effects of ASX (Table 1). Regarding exercise intervention, most studies implemented a moderate- or high-intensity exercise protocol, with the exception of one mouse study, which showed that low-intensity exercise, associated with ASX intake, improves cognitive function [36].

Among all studies, only three were able to investigate the interaction between exercise and ASX supplementation [34,39,42]. One study found additive effects on exercise performance. The synergistic effect could not be evaluated since there were no data for ASX intervention alone. The additive effect of ASX on exercise training was explained by describing the improving fat metabolism and decreasing oxidative stress [34]. A second interesting article found two distinct interaction effects (antagonistic and additive); however, there was no ASX intervention alone to measure the synergistic effect. Astaxanthin exhibited antagonistic interaction with exercise, with regard to antioxidant enzyme activity. In other words, ASX suppressed the exercise-induced enhancement of antioxidant enzymes. However, ASX exhibited an additive effect by decreasing oxidative stress and ameliorating the nitric oxide system [39]. These findings call attention to the fact that ASX can interact with exercise in opposite ways, depending on which physiological variables are being analyzed. Finally, a third study found an additive, but not synergic, effect of ASX: it led to increased newborn neurons [42]. In summary, ASX supplementation seems to interact with exercise regimes in an additive manner by improving metabolism, redox status, and neurogenesis. A more well-controlled study design is needed to determine whether ASX and exercise exhibit synergic effects.

As shown in Table 1, the most popular examination of ASX associated with exercise in animal studies is its potential to increase exercise performance. The idea that ASX may improve exercise performance is not recent. The first report published is dated 15 years ago, in 2006 [26]. The reason for this hypothesis comes from the fact that ASX may delay exercise fatigue by improving lipid metabolism via the enhancement of peroxisome proliferator-activated receptor-γ coactivator (PGC-1α) [28]; it also increases mitochondrial biogenesis via AMP-activated protein kinase (AMPK) pathways [33], and it may accelerate muscle recovery through the amelioration of redox status [32].

Astaxanthin intake has been tested for many diseases in which oxidative stress and inflammation play a critical role in pathophysiological progressions, such as diabetes, cardiovascular disease, cancer, and dementia [43]. Dementia is one group of diseases for which inflammation and oxidative stress are key developmental factors that lead to further cognitive decline [44]. A recent study found preventive effects of ASX in a humanized APP mice model of Alzheimer’s disease (AD) [45]. In this study, the authors mixed ASX into the animals’ diet from the juvenile period. This mouse model carries familial AD mutations (Swedish, Beyreuther/Iberian, and Arctic) via knock-in techniques, and it is expected to exhibit AD pathology hallmarks starting from three months of age [46]. The ASX group exhibited decreased AD onsets compared to the control group after nine months [45]. Interestingly, a rat study [47] and a monkey study [48] found ASX in brain tissues, in specific areas such as the cortex and hippocampus, suggesting that this compound can cross the blood–brain barrier and may exert a direct effect on cognitive function.

Exercise has been recognized as a non-pharmacological intervention for the treatment or prevention of dementia [49]. These neuronal benefits of exercise on brain function can be explained through several molecular mechanisms, for example via Irisin-myokine release [50], via neurotrophic factors such as brain-derived neurotrophic factor (BDNF) [51] or insulin-like growth factor 1 (IGF-1) [52], by modulating dopamine turnover [53], or by improving the cardiovascular system [54]. On the other hand, many nutrients have been indicated as potential neuromodulators that can regulate synaptic plasticity and thereby improve cognitive capacity [55]. However, little is known about how dietary nutrients can increase the well-known positive benefits of exercise on cognitive function.

Surprisingly, one animal study suggests that ASX intake further increases exercise-induced neurogenesis [42] (Figure 3). The role of newborn neurons in cognitive function is still under debate, but there is consensus that this phenomenon contributes to neuroplasticity, memory, and learning [56]. Furthermore, this same report suggests that the additive effects of exercise and ASX on enhancing neurogenesis are mediated by brain-derived leptin signaling [42]. Leptin is mainly known as a hormone responsible for controlling food intake, but its role on cognitive function and cognitive decline has also been extensively investigated [57]. The main molecular mechanism of leptin’s effects on learning and memory can be explained by its capacity to increase hippocampal synapses and plasticity [57]. Although more studies are necessary to confirm the effects of ASX on leptin metabolism, the effects of exercise training on leptin are already well established [58]. Therefore, improvement of cognitive function might be a promising research field to evaluate the additive or synergistic effects of ASX and exercise.

## 4. Astaxanthin and Exercise: Human Studies

Although animal study results are promising regarding the ergogenic effects of ASX in sport science, human research data are controversial. Among the human studies examined for the present review, there are only six studies about ASX supplementation and exercise (Table 2), all of which tested its ergogenic effects in non-sedentary individuals. Three revealed no performance improvement with ASX [59,60,61], two did reveal performance improvement (cycling time-trial test) [62,63], and the final one revealed only an antioxidant effect [64]. Interestingly, a low dosage (4–12 mg/day) appears to be more promising than a higher dosage (20 mg/day). The duration of ASX supplementation was between one and twelve weeks. Interestingly, one study suggests that a short duration (one week) is enough to increase the time to exhaustion [63]. It is worthy of note that all ASX used in the human studies presented here were from natural sources and not synthetic. However, due to the low number of studies, at present, it is not possible to draw conclusions about the ergogenic effects of ASX in humans.

As expected, animal studies have revealed more promising results. This is common in translation studies because the number of possible confounding factors is reduced in animal research. Another explanation may be due to the fitness level of participants in the research presented here. All studies tested ASX supplementation in non-sedentary subjects engaged in high-level protocol training. Therefore, ASX supplementation in sedentary subjects engaged in low to moderate training protocols should be tested.

The weak evidence for the effects of ASX supplementation in human studies may be partially explained by ASX’s bioavailability. Generally, carotenoids have low bioavailability, and this is also true for ASX [65,66,67,68,69,70]. The absorption, distribution, metabolism, and excretion of ASX are mainly reported in animal studies rather than human studies [71]. Since ASX bioavailability differs between rats and humans [72], it is difficult to extrapolate the main existing data to clinical trials. Moreover, human studies are controversial since many factors can influence ASX bioavailability, such as oil intake [73], ASX esterification [66], diet, and smoking habits [74]. Therefore, more human studies are necessary to completely understand ASX dynamics and kinetics in order to establish the optimal intake or supplement dosage.

Another possible reason for the lack of evidence in human studies is linked to the redox status dynamic of exercise physiology. Robust evidence emphasizes the importance of exercise-induced oxidative stress as a molecular signal trigger to promote long-term exercise adaptations [75]. Furthermore, the attenuation of this oxidative stress stimulus via antioxidant supplementation seems to mitigate exercise performance improvements. Therefore, it is expected that a strong antioxidant approach in the sports field should be undertaken with caution [75].

There were no human studies found that evaluated the effect of ASX supplementation alone or that had a sedentary control group. Consequently, it is difficult to analyze the possible interactions between the two interventions. Nevertheless, it is premature to assume that ASX and exercise do not interact in humans. First, many studies have been performed with the goal of increasing ASX bioavailability [65,66,67,68,69,70]. Therefore, it is expected that in the future new ASX formulations will provide better outcomes. Second, ASX and exercise paradigms should be also be tested under different conditions rather than exercise performance only: for instance, cognitive function. Third, as mentioned above, the non-trained individual may exhibit a strong response to a training and supplementation configuration. For these reasons, the scientific community should continue performing clinical trials in order to increase evidence for the combined effect of ASX and exercise in humans.

## 5. Overview of the Current Scientific Literature

Exercise training and ASX intake promote similar physiological modulations that induce the hypothesis of an additive or synergistic effect when both are applied simultaneously. For instance, ASX intake and exercise are potential non-pharmacological interventions for the prevention or treatment of several chronic non-communicable diseases, from metabolic diseases to neurodegenerative diseases [15,76]. Moreover, basic studies point to ASX as an ergogenic supplement that enhances exercise performance [77], though positive results in human studies remain scarce [75]. However, a more prominent additive effect between exercise and ASX might be related to cognitive domains. The beneficial effects of exercise on cognitive function are well known [78], and many molecular mechanisms have emerged to explain this phenomenon. On the other hand, ASX’s effects on the brain have also been targeted for investigation due to its neuroprotective characteristics [77]. Recently, a large study investigating a multifactorial approach to the prevention of dementia was performed. To summarize the study’s results, targeting more than one risk factor simultaneously seems to be the best way to fight cognitive decline [79]. Therefore, future research should make an effort to understand how multiple interventions work in an additive or synergic manner that strengthens the potential therapeutic effects of each individual lifestyle factor alone.

In the current scientific literature, several papers are related somewhat to ASX and exercise regimes. The main purpose of this review was to investigate the potential additive or synergistic effects of both interventions, given their similar primary outcomes. Unfortunately, studies that investigated potential interactions are scarce. The difficulty of analyzing the combination of both intervention effects may partially explain this lack of evidence. To explore this, researchers should first establish the effects of each intervention alone; after that, it is necessary to find a statistical difference between the groups for which both interventions are simultaneously applied. Another challenge is that the synergistic effects must statistically overcome the strong effects of exercise training alone.

Four extensive reviews about ASX have been published in the online database. Although all of these reviews compiled excellent original investigations of ASX effects in different fields of study, the authors always concluded that more investigation was needed to ensure the best ASX supplementation protocol needed in order to achieve the most efficient effects. However, the more optimistic conclusion about ASX supplementation among the reviews is regarding its neuroprotective effects (Table 3).

## 6. Conclusions and Perspectives

Overall, the majority of studies have tested the effect of ASX supplementation on exercise performance parameters. No synergistic effects have been found. However, the most promising effect of ASX and exercise interacting in an additive manner might be related to cognitive function. This new finding may open up an original area of research: the additive physiological function of nutrition and exercise for improving brain function. Unfortunately, little is known regarding the interaction of ASX and exercise for many of the other well-known physiological changes caused by both interventions separately (Figure 4). Therefore, more studies investigating exercise intervention plus ASX intake are warranted.

## Figures and Tables

**Figure 1 antioxidants-10-00870-f001:**
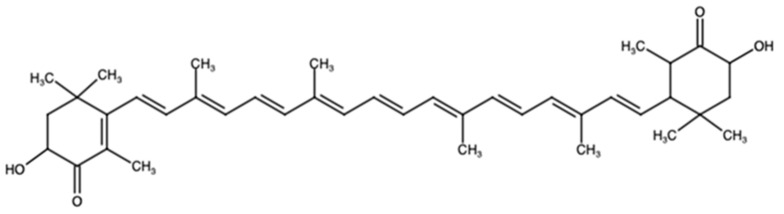
Astaxanthin’s non-stereochemical structural formula.

**Figure 2 antioxidants-10-00870-f002:**
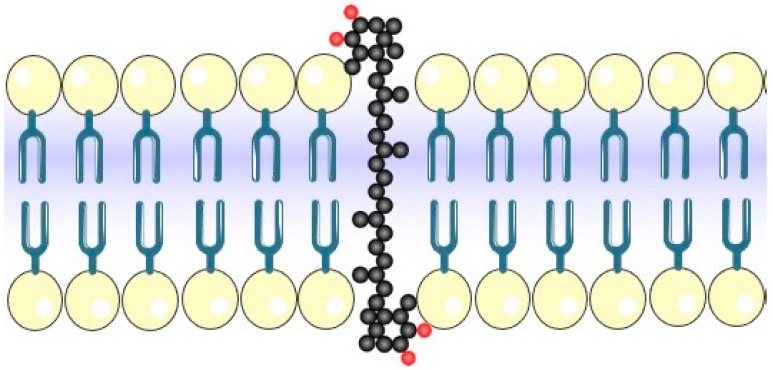
Astaxanthin cell-membrane insertion. Carbon (black circles). Oxygen (red circles). Polar region of bilayer phospholipid membrane (yellow circles). (Illustrations provided by Servier Medical Art, licensed under the Creative Commons Attribution 3.0 unported license (https://smart.servier.com accessed in 19 May 2021)).

**Figure 3 antioxidants-10-00870-f003:**
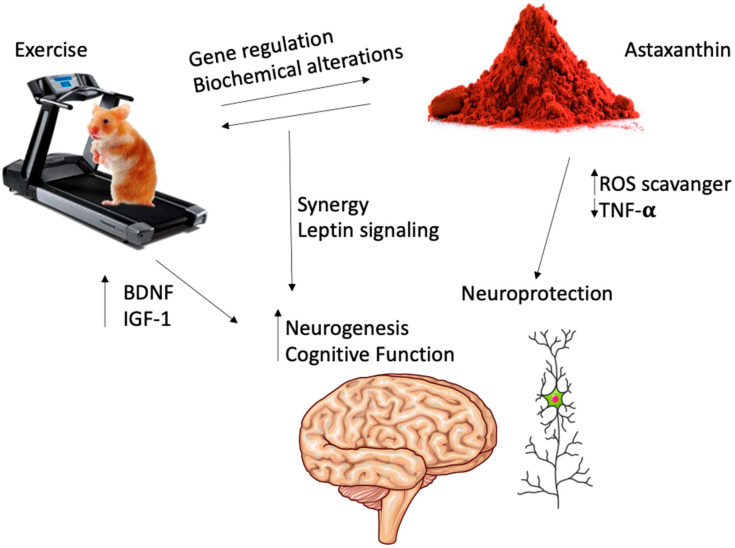
Astaxanthin and exercise modulating cognitive function. ROS: reactive oxygen species. TNF-α: tumor necrosis factor-alpha. BDNF: brain-derived neurotrophic factor. IGF-1: insulin-like growth factor one. (Illustrations provided by Servier Medical Art, licensed under Creative Commons Attribution 3.0 unported license (https://smart.servier.com accessed in 19 May 2021)).

**Figure 4 antioxidants-10-00870-f004:**
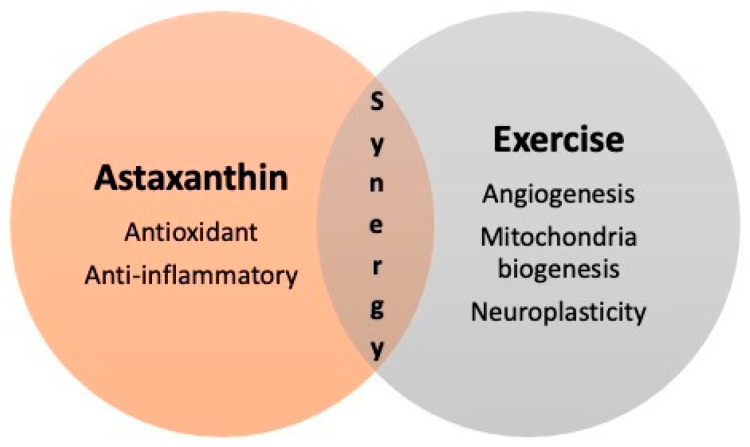
Potential additive or synergistic effects between ASX and exercise.

**Table 1 antioxidants-10-00870-t001:** Animal studies.

Author (Year)	Animal (Age)	Intervention (Duration)	Primary Outcome	Results
M. Ikeuchi (2006)	ddY mice (4 weeks)	ASX: 30 mg/kg b.w. (gavage) Exe: swimming to exhaustion (5 weeks)	Endurance performance	Increased time to exhaustion by increasing lipid metabolism [33]
**W. Aoi ** **(2008)**	**ICR mice** **(7 weeks)**	**ASX: 0.02% *w*/*w*** **Exe: 2/wk, 18 m/min—5 min** **(4 weeks)**	**Endurance** **performance**	**Increased time to exhaustion by** **increasing lipid metabolism** [34]
H. Liu (2014)	ICR mice (7 weeks)	ASX: 0.02% *w*/*w* Exe: 30 min 25 m/min treadmill (2 weeks)	Lipid metabolism	Increased PGC-1alpha in skeletal muscle [38]
T. G. Polotow (2014)	Wistar rats (NA)	ASX: 1 mg/kg bw (gavage) Exe: swimming to exhaustion (6 weeks)	Endurance performance	Increased time to exhaustion by redox balance [35]
T. Shibaguchi (2016)	Wistar rats (14 weeks)	ASX: 0.04% *w*/*w* (6 weeks)	Muscle atrophy	Attenuated skeletal muscle atrophy by redox balance [37]
W. Aoi (2017)	ICR mice (8 weeks)	ASX: 0.02% *w*/*w* Exe: 3/wk, 25 m/min—5 min (5 weeks)	Endurance performance	Increased time to exhaustion [36]
**Y Zhou** **(2019)**	**C57BL/6J mice** **(7 weeks)**	**ASX: 30 mg/kg bw (gavage)** **Exe: 45 min moderate** **swimming** **(4 weeks)**	**Redox status**	**Suppressed antioxidant enzyme activity** [39]
Y. Nishida (2020)	C57BL/6J mice (6 weeks)	ASX: 0.02% *w*/*w* (24 weeks)	Insulin resistance	Increased mitochondria biogenesis via AMPK pathway [40]
J. S. Yook (2016)	C57BL/6J mice (11 weeks)	ASX: 0.5% *w*/*w* (4 weeks)	Cognitive function	Increased spatial memory by increasing hippocampal neurogenesis [41]
**J. S. Yook** **(2019)**	**C57BL/6J mice** **(11 weeks)**	**ASX: 0.5% *w*/*w*** ** Exe: mild treadmill running** **(4 weeks)**	**Cognitive** **function**	**Increased spatial memory with** **increasing hippocampal neurogenesis** [42]

ASX: Astaxanthin, Exe: exercise, b.w.: body weight, NA: not applicable. Bold font indicates studies in which additive effects between ASX and EXE were found.

**Table 2 antioxidants-10-00870-t002:** Human studies.

Author (Year)	Subjects	Intervention	Primary Outcome	Results
R.J. Bloomer (2005)	Resistance-trained males	4 mg/d, 3 weeks	Muscle performance	No difference [59]
C. P. Earnest (2011)	Amateur endurance-trained males	4 mg/d, 4 weeks	TT performance	Improved performance [62]
B. Djordjevic (2012)	Male elite soccer players	4 mg/d, 12 weeks	Redox status	Stress oxidative prevention [64]
P.T. Res (2012)	Well-trained male cyclists or triathletes	20 mg/d, 4 weeks	TT performance	No effect [60]
L.J.J. Klinkenberg (2013)	Well-trained male cyclists	20 mg/d, 4 weeks	Redox status	No effect [61]
D.R. Brown (2021)	Trained male cyclists	12 mg/d, 1 weeks	TT performance	Improved performance [63]

TT = time trial.

**Table 3 antioxidants-10-00870-t003:** Review studies.

Author (Year)	Purpose of the Study	Practical Value
S. A. Mason (2020)	The authors performed an extensive and critical literature review regarding the most common antioxidant supplements for athletes (e.g., ASX, catechins, curcumin, quercetin, resveratrol, vitamin C., etc.)	After considering ASX evidence, the authors concluded that there is a lack of evidence to support it as a supplement [75]
D. R. Brown (2018)	In this review, the authors rigorously interpreted scientific literature regarding the ergogenic effects of ASX. Although they recognized the promising data found in in vitro and in vivo research, human studies were not satisfactory.	Their final message was that future investigation is needed regarding ASX ergogenic effects in humans [31].
B. Grimmig (2017)	In this narrative review, the authors introduced a wide range of evidence from basic studies to clinical trials for the possible effects of ASX on cognitive function.	After the discussion, the authors conclude that ASX is a promising therapeutic agent for neurodegenerative diseases [77].
J.P. Yaun (2011)	Using a broad approach, ASX’s biological effects were reviewed. The authors summarized the scientific evidence for ASX’s effect on several disease conditions.	Finally, the authors stated that although it is encouraging, more extensive and well-controlled clinical trials are necessary [15]

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
