# Peer review of "Benefits of Exercise and Astaxanthin Supplementation: Are There Additive or Synergistic Effects?"

_antioxidants, 2021, doi:10.3390/antiox10060870_

Round 1

Reviewer 1 Report

This is a review paper on the impact of astaxanthin and exercise in cognitive functions.

The paper is well written.

Some points to consider are these:

  1. figure 3 is not clear, especially at the bottom of the figure, it is not clear whether the neurogenesis and cognitive function are coming from. The figure needs to be edited so it is easier for the reader to follow.
  2. the anti-inflammatory properties of astaxanthin and/or exercise are not extensively reviewed. I would invite the authors to a make a link between astaxanthin and inflammation and also to inflammation related diseases, such as cancer, CVD, stroke and diabetes type 2.
  3. Table 3 needs to be improved, I would suggest the authors to "break" the highlights column to 2-3 columns mentioning a. the type of the study, b. the impact of the examined parameter and c. its practical value.

I am happy to review the revised version of the MS.

Author Response

Reviewer 1

1. figure 3 is not clear, especially at the bottom of the figure, it is not clear whether the neurogenesis and cognitive function are coming from. The figure needs to be edited so it is easier for the reader to follow.

Improvement was done - Leandro

2. the anti-inflammatory properties of astaxanthin and/or exercise are not extensively reviewed. I would invite the authors to a make a link between astaxanthin and inflammation and also to inflammation related diseases, such as cancer, CVD, stroke and diabetes type 2.

Invitation accepted - Leandro

3. Table 3 needs to be improved, I would suggest the authors to "break" the highlights column to 2-3 columns mentioning a. the type of the study, b. the impact of the examined parameter and c. its practical value.

Done - Leandro

Reviewer 2 Report

The present review paper is focused on astaxanthin, a red carotenoid pigment occurring in seafood, which has been recently recognized in the scientific literature as a potential nutraceutical due to its antioxidant, anti-inflammatory, and neurotrophic properties. Specifically, the main scientific evidences for the astaxanthin supplementation in association with exercise regimes were studied.

The review is interesting and emphasizes the importance on the combination of exercise and nutritional supplementation in a view of future research directions in the field of sports nutrition science.

I do have some remarks.

  • English requires revision. Some misspellings can be found throughout the whole text e.g. line 13, “healthy”.
  • Line 60. It would be interesting to mention the other languages of the papers appeared in the PubMed online database.
  • Section 5. Overview of the current scientific literature. This section needs to be improved. The authors mention four extensive reviews about astaxanthin published in the online database. Why did not the authors report original works as done for Sections 3 and 4.

Author Response

Reviewer 2

  • English requires revision. Some misspellings can be found throughout the whole text e.g. line 13, “healthy”.

English writing was improved. Alterations were tracked in the word file. Leandro

  • Line 60. It would be interesting to mention the other languages of the papers appeared in the PubMed online database.

Only English papers were found. Leandro

  • Section 5. Overview of the current scientific literature. This section needs to be improved. The authors mention four extensive reviews about astaxanthin published in the online database. Why did not the authors report original works as done for Sections 3 and 4.

The authors did not mention original works in section five because, as reviewer 2 pointed out, we did it in sections 3 and 4. Since scientific writing must be concise, succinct, precise, objective, and without unnecessary repetition of information, section 5 was focused on review papers. Leandro

Round 2

Reviewer 1 Report

The paper can now be accepted, as it is.

I would like to thank the authors for their replies.

Reviewer 2 Report

The paper has been improved in comparison with its first submission and the paper can be now accepted in the form as it is.